# Adopting sustainable innovations for remote access to TB and HIV care in South Africa

**Michael Galvin** [1,2,3]*, **Denise Evans** [3]*, **Aneesa Moolla**[3], **Lezanie Coetzee**[3], **Vongani Maluleke**[3], **Patricia Leshabana**[3], **Jacqui Miot**[3]

**1** Department of Psychiatry, Boston Medical Center (BMC), Boston, Massachusetts, United States of America, **2** Harvard T.H. Chan School of Public Health, Boston, Massachusetts, United States of America, **3** Health Economics and Epidemiology Research Office, Faculty of Health Sciences, University of the Witwatersrand, Johannesburg, South Africa

* michaelgalvin@wustl.edu (MG); devans@heroza.org (DE)

**Data Availability Statement:** We have provided the raw data as a "Supplementary File". This submission contains all raw data required to

## Abstract

For the last decade, South Africa has made substantial progress to control the dual HIV and TB epidemics. However, disruptions in TB and HIV treatment during the COVID-19 pandemic threatened to reverse this. This study aimed to identify adaptations in HIV and TB service delivery models in response to COVID-19 and government restrictions. This information informed the development of an online survey, which was utilized as part of a consultation exercise to further capture adaptations made to HIV/TB service delivery within the South African context. The literature review involved screening 380 titles and abstracts, identifying 30 HIV and TB studies across 19 countries, and categorizing 90 individual interventions into ten thematic areas. Common themes included interventions addressing screening, testing, diagnosis, medication collection and delivery support, and virtual models. Digital health interventions and adaptations to medication collection/delivery were reported in 38% of studies. Analysis of survey responses from 33 stakeholders in South Africa revealed that 47% of interventions targeted HIV, 11% TB, and 23% HIV/TB integrated service delivery. Most interventions (81%) were integrated into the national HIV or TB program, with implementation occurring at various levels: 39% at facility level, 35% at sub-district or district level, and 18% at provincial level. Programmatic data was available for 86% of interventions, with 50% being funded. This study demonstrated that services can be delivered in locations other than in health facilities (e.g., community-based or home-based) and that integrated services can also free up additional resources. Although studies varied, COVID-19 accelerated the adoption of differentiated service delivery (DSD) models for TB care, including multi-month dispensing (MMD) for TB preventative therapy (TPT) and TB treatment, home-based or mobile outreach screening and testing, and community pickup points (PuP) for TB medications. These initiatives had previously lagged behind HIV-focused DSD models, and it is crucial to sustain these services beyond the pandemic. To achieve universal health coverage, it will also be important to capitalize on these experiences and learn from HIV-focused DSD models so programs can deliver integrated person-centered chronic care services for TB, HIV, and non-communicable diseases.

replicate the results of our study. Data can also be available upon request from the authors or from the Health Economic & Epidemiology Research Office (HE2RO) at the University of the Witwatersrand which can be reached at information@heroza.org.

**Funding:** This work has been made possible by the generous support of the American People and the President's Emergency Plan for AIDS Relief (PEPFAR) through the United States Agency for International Development (USAID) under the terms of Cooperative Agreement 72067419CA00004 to HE2RO. The contents are the responsibility of the authors and do not necessarily reflect the views of PEPFAR, USAID or the United States Government. https://www.state.gov/pepfar/ The funders had no role in study design, data collection and analysis, decision to publish, or preparation of the manuscript.

**Competing interests:** The authors have declared that no competing interests exist.

# Background

In January 2020, China announced a cluster of cases referred to as COVID-19 and in just two months the WHO declared COVID-19 a pandemic [1]. By 1 March 2020, total confirmed COVID-19 cases in China were between 80 000–84 000 with between 3 000–4 640 deaths [2]. Globally, the total confirmed cases surpassed 100,000. In March of the following year, over 113 million cases had been confirmed worldwide with 1.98 million COVID-19 deaths reported [3]. The pandemic caused intense uneasiness and concern across the world causing countries to close their borders as part of implementing disease control strategies to minimize the risk of infection [4].

At the same time, to reduce transmission of COVID-19, South Africa declared a State of Disaster and implemented a strict national lockdown where only essential services were permitted [5]. For the remainder of 2020 and 2021, varying levels of restrictions were adopted in response to the re-emergence of COVID-19 in the months that followed [6].

Routine healthcare services in South Africa, which already has a high burden of TB and HIV, were negatively impacted during the pandemic [7–10]. Globally in 2020, 37.7 million people were living with HIV (PLHIV), with 20.6 million residing in East and Southern Africa alone [11, 12]. Using national, public sector, and facility-level data, Benade and colleagues showed that ART initiations dropped by 28% in 2020 compared to the previous year, with District hospitals, larger facilities, and urban areas experiencing the largest reductions [9]. In the 2021 Global TB Report, the WHO reported an increase in TB cases and deaths from TB for the first time in decades. Fewer cases of TB were detected and fewer people were treated for TB during 2020/2021 due to disruptions caused by COVID-19. Reduced access to TB testing and treatment resulted in an increase in TB deaths, back to the level of 2017, along with an increase in TB incidence. Other impacts included reductions in the number of people provided with treatment for drug-resistant TB and TB preventive treatment, and a fall in global spending on TB diagnostic, treatment, and prevention services.

Literature has highlighted the potential impact of the pandemic and lockdown on patients and healthcare services in South Africa [13–15]. Factors such as fear of contracting COVID-19 and associated stigma [16, 17], restrictions on movement, increased family responsibilities, unreliable transport, and loss of income or employment negatively impacted health seeking behavior and access to services [18]. In terms of services, the deprioritization of routine health services, diversion of resources, and repurposing of the health workforce led to the reordering of healthcare priorities from both the demand and supply sides [19].

Because these disruptions threatened to overturn gains made by the HIV and TB programs in South Africa, activities to address the unintended consequences of the pandemic had to be prioritized by all sectors [11]. However, few reports described how service delivery was adapted during the pandemic [20–22]. To better understand how TB and HIV service delivery was adapted in response to COVID-19 and the government restrictions imposed, we conducted a review of available literature as an initial step before conducting a study that reviewed interventions specific to South Africa through a comprehensive online survey with healthcare experts throughout the country.

# Methods

## Literature review

A rapid literature review was conducted to document how HIV and TB service delivery models had been adapted in response to the COVID-19 pandemic and government restrictions. We reviewed abstracts and publications between March 2020 and December 2021 which described

**Table 1. Summary of the review according to the PRISMA checklist.**

| | Inclusion | Exclusion |
|---|---|---|
| Rationale | Understand how HIV and TB service delivery models were adapted during COVID-19 pandemic and government restrictions | |
| Objective | Review available literature to identify key intervention groups/themes | |
| Eligibility criteria | Inclusion | Exclusion |
| Population | Human patients/people/children/adolescents with TB) who have HIV or TB. All ages. All genders | Non-human studies i.e., animal |
| Intervention | Health system interventions focused on how HIV and/or TB services are delivered only (e.g. service delivery, health workforce, and health information systems only) | Observational studies with no intervention |
| Outcomes | Reporting on the impact of the intervention (e.g. efficacy or success rate), and other process indicators for implementations (feasibility, acceptability). | N/A |
| Timing | Studies published from March 2020 and December 2021 | N/A |
| Setting | All countries<br>All levels of the health system (primary, secondary, tertiary, and quaternary care) and community care | N/A |
| Study design | Intervention studies, including:<br>• Intervention trials<br>• Cross-sectional studies<br>• Pre/post studies with or without a comparison group<br>• Descriptive studies with individual patient data or health provider information<br>• Qualitative and quantitative studies | Case reports/case series<br>Mathematical modelling studies<br>Pharmacokinetic or toxicodynamic models<br>Conference Proceedings<br>Basic science articles, focused on mechanisms of TB disease or treatment. |
| Publication type | Articles published in peer-reviewed, scientific databases as listed.<br>English language only | Editorials, letters, commentaries |
| Information sources | MEDLINE and websites (IAS and WHO) | |

"programmatic innovations" or "differentiated service delivery" to address service delivery during COVID-19. During this period, 273 million COVID-19 cases and over 5.3 million deaths were reported globally while in South Africa ~3 million total COVID-19 cases were reported by December 2021. The literature search was conducted between May and August 2022.

Two independent reviewers (DE and VM) screened the records and retrieved the data. Full-text reviews were conducted by two independent reviewers, with conflicts resolved by discussion. After we removed duplicates, data for full English language publications were captured into an Excel spreadsheet template. We extracted data on the intervention type (description), intervention period, sample size, setting/site, outcome (efficacy/impact), success rate, and other process indicators for implementations (feasibility, acceptability). We reviewed the extracted data and grouped interventions into key intervention groups or themes (S1 Table). As this was a rapid review, we undertook a descriptive analysis only, and study quality was not formally assessed. A summary of the review according to PRISMA guidelines and search strategy in MEDLINE can be seen in Tables 1 and 2.

## South Africa study

Between June 1st and August 31st 2022, we conducted a rapid electronic survey to document changes or innovations that were implemented in South Africa to address disruptions in service delivery as a result of COVID-19. A comprehensive online survey was designed to capture a range of information including (1) a description of any changes/adaptations/innovations to HIV or TB service delivery in response to the COVID-19 pandemic, (2) the setting where implemented, (3) population and sample size, (4) duration and sustainability of the intervention, (5) funding and (6) the nature and availability of any data. We asked specifically about the key intervention groups/themes that we identified in the rapid literature review.

**Table 2. Search strategy in MEDLINE.**

| Area | Search terms |
|---|---|
| COVID-19 | (coronavirus* or 2019nCoV* or 19nCoV* or "2019 novel*" or Ncov* or "n-cov" or "SARSCoV-2*" or "SARSCoV-2*" or SARSCoV2* or "SARS-CoV2*" or "severe acute respiratory syndrome*" or COVID*2).ti,ab,kw,kf. exp COVID-19/ or exp Severe Acute Respiratory Syndrome/ or exp SARS-CoV-2/ or exp Severe acute respiratory syndrome-related coronavirus/ |
| Government restrictions | (Lockdown* or lock down* or (government adj restrict*) or ((epidemic? or pandemic* or global* or international or worldwide or world wide or national or regional or mass or population* or impose? or imposing or enforc* or forc* or mandat* or voluntary or polic*) adj5 (quarantin* or isolat*)) or ((travel* or movement) adj1 restrict*) or (clos* adj2 border*) or "flatten* the curve").ti,ab,kw,kf.<br>exp quarantine/ or exp communicable disease control/ |
| Tuberculosis | (Tuberculo* or tubercular or koch* disease or Mycobac*).ti,ab,kw,kf. or exp tuberculosis/ or exp Mycobacterium tuberculosis/ |
| HIV | (Human Immunodeficiency Virus* or Immunodeficiency Virus* or Acquired Immun* Deficiency Syndrome or Acquired Immunodeficiency Syndrome or Acquired Immunodeficiency Syndrome or Acquired Immun* Deficiency virus* or Acquired Immunodeficiency virus* or Acquired Immunodeficiency virus* or AIDS syndrome* or AIDS Virus* or HIV).ti,ab,kw,kf.<br>Or exp hiv/ or exp Acquired Immune Deficiency Syndrome/ |
| Study restrictions | |
| Date | limit to yr = "2020–2021" |
| Category | |
| Service delivery | (deliver* or adapt* or implement* or mitigat* or innovat* or access* or system* or program* or distribut* or staff* or workforce* or personnel* or maintain* or delegat* or practic*).ti,ab,kw,kf.<br>exp "Delivery of Health Care"/ or exp Health Services/ or exp workforce/ or exp health workforce/ or exp health personnel/ |

* ti,ab,kw,kf (Title, Abstract, keyword field, and keyword heading field); exp ("exploding" the subject heading; i.e., any terms beneath main subject heading)

The online survey was sent out to local stakeholders, organizations/facilities/implementing partners, or specific healthcare departments that provide HIV and/or TB services. District support partners (DSPs), HIV and TB forums, the National Department of Health (NDoH) HIV and TB Think Tanks, non-governmental organizations, and all other healthcare or health systems strengthening organizations were also e-mailed. The cross-sectional survey was developed and piloted with a small group of stakeholders (n = 5) before the survey was finalized and e-mailed to over 240 stakeholders (S2 Table). We used snowball sampling to reach additional stakeholders.

The primary target population included NDoH-employed staff at the national, provincial, and district levels, as well as partners or organizations who implemented or supported changes, adaptations, and innovations to HIV or TB service delivery during COVID-19. Inclusion criteria included: (1) staff working in HIV or TB service delivery since March 2020 (2) willingness to participate and provide consent (3) having sufficient time to participate and (4) being 18 years or older.

An invitation to participate was sent out as described above with a link to an online form. Once respondents confirmed their eligibility and provided consent, a unique study ID was assigned. The survey data was collected in REDCap, an electronic data capture tool hosted by the University of the Witwatersrand [23, 24]. Once a study ID was assigned, respondents could add their responses to the survey. The e-survey contained 32 questions related to changes, adaptations, and innovations to HIV or TB service delivery during COVID-19. At the

end of the survey, respondents were offered either a small reimbursement for their time (ZAR50; equivalent to USD $2.6) or an opportunity to donate this amount to a charitable organization. The survey took approximately 10 minutes to complete but could have taken longer if respondents chose to include a detailed response.

Personal information (name, surname, email address, contact number) was kept separate from participants' responses (survey) and all data was saved on a secure server with access limited to the research team. A linking file with the personal information of participants with the assigned unique study ID was created so that participants could be contacted at a later stage in the study. The linking file was a password-protected Excel file that was stored on HE²RO's secure network, and access was restricted to HE²RO study staff only. Respondents were identified by their unique study ID in the survey data, and personal information was not included in the analytic dataset. Survey data was exported from REDCap and imported and analyzed using MS Excel and STATA software, Version 14.

This study was approved by the Human Ethics Research Committee of the University of the Witwatersrand (protocol M211198). All respondents provided written informed consent to participate in the online survey.

## Results

### Literature review

We screened 380 abstracts and titles and identified 30 HIV and TB studies conducted in 19 countries. The majority of studies were obtained from three reports which included the World Health Organization report on programmatic changes to TB services [3], an IAS supplement on how COVID-19 expedited differentiated service delivery for HIV [11] and the South African Health Review 2021. We identified 90 individual interventions which were subsequently grouped into ten intervention groups/themes (Fig 1).

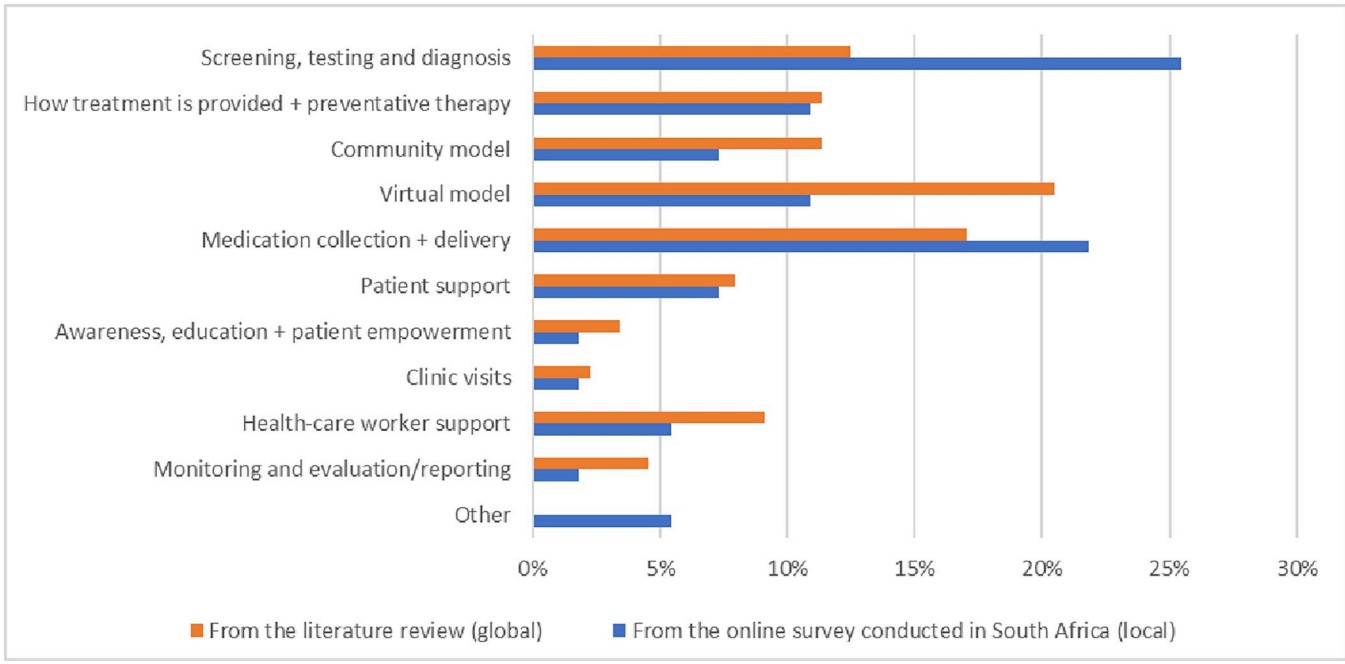

**Fig 1. Adaptations to HIV and/or TB service delivery in response to COVID-19 disruptions, grouped by key intervention groups/themes and compared (global versus local).**

Many of the countries that implemented TB interventions were on the WHO's list of high-TB or TB/HIV burden countries (e.g., India, Philippines, Pakistan, South Africa, Russian Federation, Zambia, Mozambique, amongst others.). Studies varied by geographic location, sample size, scope, and measurement of outcome, so it was difficult to compare practical, feasible, and acceptable interventions. No studies reported cost-effectiveness and few reported patient/provider experiences. Challenges in implementation, such as lack of support from healthcare workers, stigma, difficulty accessing telephone services for telemedicine, and professional burnout, were well documented.

Interventions that addressed adaptations to screening, testing, and diagnosis, supported medication collection and delivery and virtual models were the most common intervention groups/themes (Fig 2). Digital health interventions and adaptations to medication collection or delivery were described in 33 (38%) reports. Changes to clinic visits, monitoring and evaluation/reporting, healthcare worker support and patient education, awareness, and empowerment were the least explored interventions.

While COVID-19 affected every facet of the health care delivery system, including the workforce, some suggest that rural areas appeared to be more resilient than urban areas and that larger urban and peri-urban areas saw greater reductions in treatment initiations that rural areas [10]. The review showed that services could be delivered in locations other than in health facilities, closer to the patient's home. Home-based (e.g., doorstep diagnostic services, home delivery of medication) or community-based services (e.g., self-testing at community check points, mobile outreach with digital chest X-rays, health promotion messages, peer led counselling or village support groups, community delivery/collection of medication etc.) offered several benefits to patients including fewer health facility visits, improved accessibility, less travel, less expensive, convenience and reduced risk of infections.

While some changes may have been made in response to challenges specific to the setting–due to stay-at-home orders, restrictions on transport, reallocation of health care capacity or drug shortages–for example–lessons learned can inform which interventions could be scaled up or integrated into routine care to provide high-quality person-centered care for HIV and/or TB. Our literature review revealed gaps in the perspective and experience of both recipients of care and healthcare providers of COVID-19 adaptations to HIV and TB service delivery, as well as cost data that can be particularly relevant in making the argument for adaptations to be sustained leading to a better service delivery system going forward. Evidence is also needed to understand the effects and challenges of the temporary measures implemented to bridge TB and HIV service delivery during COVID-19.

## South Africa study

We received responses to the online survey from seven of the nine provinces in South Africa. In total, 36 researchers, government employees (e.g., from NDoH and medical officers at government facilities), laboratory personnel, medical doctors, implementing partners, and members from the South African National AIDS Council (SANAC) responded to the survey. After removing three organizations that did not implement any service delivery interventions, 33 respondents and 57 independent interventions, were included in the analysis. 21% (7/33) of respondents were from the public or government sector while 12% (4/33) were from private, 48% from non-governmental (16/33), and the remainder from other organizations (6/33; 18%). Among the interventions reported; 47% (27/57), 11% (6/57), and 28% (16/57) targeted HIV, TB, and HIV/TB integrated service delivery, respectively (8/57 not stated).

In terms of implementation, 39% (22/57) of interventions were implemented at the facility, 35% (20/57) at the sub-district or district level, and 18% (10/57) at the provincial level. At the

**Fig 2. Interventions grouped across the care cascade.**

time of the e-survey, 81% (42/52) of the interventions had been integrated as part of the national TB or HIV program or as the standard of care. Among respondents who reported collecting data, data was available without restrictions for 86% (31/36) of the interventions.

**Table 3. Summary of responses, by place where the interventions were implemented.**

| Setting | Sample size (range) | Patient population* | % of interventions integrated into routine care [Duration of intervention in months (median IQR)] | Funding received | Availability of data and in what format (paper, electronic or both) |
|---|---|---|---|---|---|
| Facility (n = 22) | 50–3,000 | General population (1) PLHIV (10) Key populations (MSM) (5) HIV and TB (3) Adolescents living with HIV (1) Rifampicin resistant TB (1) | 20/22 (91%) [11.1 months (4.0–17.5)] | Yes 6/22 (27%) Don't know 4/ 22 (18%) | Yes 14/22 (64%) Mostly electronic (11/14) Don't know 5/22 (23%) |
| Sub-district of District (n = 20) | 1,000–300,000 | General population (6) PLHIV (5) Stable PLHIV (2) HIV and TB (2) Adolescents living with HIV (1) HIV positive young males (1) Men who have sex with men (MSM) (1) | 14/20 (70%) [22.2 months (21.4–22.9)] | Yes 9/20 (45%) Don't know 8/20 (40%) | Yes 11/18 (61%) Mostly electronic (6/11) Don't know 6/18 (33%) |
| Province (n = 10) | 200,000 –<1 Million | TB and HIV (3) TB (2) AGYM, pregnant and breast-feeding mothers (1) Key populations including sex workers (2) General population (1) | 8/10 (80%) [Not reported] | Yes 4/10 (40%) Don't know 2/10 (20%) | Yes 6/10 (60%) Paper and electronic (4/6) Don't know 3/10 (30%) |

*descending order—with the caveat that we asked specifically about changes to HIV and TB service delivery.

Abbreviations: AGYM adolescent girls and young women, MSM Men who have sex with men, TB tuberculosis, PLHIV people living with HIV

Among respondents who provided information about funding, 50% (19/38) indicated that their interventions were funded by international donor organizations such as PEPFAR, USAID, CDC, UNITAID, or local donor funders like Networking HIV and AIDS Community of Southern Africa (NACOSA) (Table 3).

The reported adaptations to HIV and/or TB services during COVID-19 were grouped into the ten intervention groups/themes. The key intervention groups/themes for the online survey results (local) were then compared to the key intervention groups/ from the rapid literature review (global) in Fig 1.

For the local service provider responses, the largest number of reported changes/adaptations were related to screening, testing, and diagnosis. The most common change in screening, testing, and diagnosis was by integrating HIV/TB testing into the COVID-19 testing and treatment pathway, with a third of the reports (14/47; 30%) indicating this change. Others reported moving more toward telephonic visits or engaging patients digitally where possible as opposed to clinic visits due to COVID-19, as well as promoting self-screening and self-testing.

The second most reported theme was related to medication collection and delivery, with more than 20% of respondents noting this as a change. The most common change in this area was by delivering medication to patients' homes or neighborhoods as opposed to requiring them to come pick up their medications. One respondent reported using CHWs and Uber courier services to deliver medication to patients during this period; drones were suggested for accessing hard-to-reach areas [25] however we only had reports that the Limpopo Health Department and Greater Tzaneen Municipality used drones to spread COVID-19 related health information to communities. Another respondent said when delivery wasn't possible,

they would ensure "contactless" medication pick up (e.g., using prescription pickup kiosks, medicine lockers services, curbside pickups, or electronic delivery of prescriptions and home delivery of medication, etc.). In addition to delivering medication to their patients, several respondents reported providing several months of medication at a time–instead of just one month–to ensure that patients did not run out of medicine due to government lockdowns. In 2020 until October 2021, there was a directive from the government for 12 month scripting. This not only facilitated more ART refills outside but led to fewer clinic consultations, reduced patient costs, and enabled 6 month multi-month dispensing for ART in South Africa. The NDoH revised the HIV adherence guidelines for differentiated care services to give patients a grace period of 14 days (originally 7 days) to return to the clinic after failing to collect their treatment at a collection point.

The consideration and development of a virtual care model was also discussed by several respondents, as some clinics were faced with creating hybrid face-to-face and telehealth options for the first time. Others noted conducting virtual trainings with staff and providing communications via Short Message Service (SMS) or WhatsApp (instant messaging and voice-over-IP service). A respondent reported staff using virtual medication monitors to support TB medication adherence for patients.

Consistent with the findings of the literature review, few interventions focused on making changes to clinic visits (e.g. operating times), monitoring and evaluation/reporting, and patient education, awareness, and empowerment.

## Interventions grouped across the care cascade

Once we assessed the adaptations, where these were taking place, by whom, and for whom, we mapped what activities were being done across the HIV and TB care cascade. This information would be useful to national programs as they consider how to roll out post-COVID-19 recovery plans aimed at finding undiagnosed people, strengthening the linkage of people diagnosed to treatment, strengthening retention in care, and strengthening prevention.

Approaches to improve screening, testing and diagnosis during COVID-19 focused on expanding new tools or equipment (e.g. additional Gene Xpert MTB/RIF or Truenat machines), supporting self-screening, self-testing, and delivery of results at the patient's home, facilitating sample transport and delivery of results, and providing support and training to health-care workers. Examples of screening for more than one condition were also evident, and some adaptations integrated screening and testing for COVID-19, TB, HIV, and other non-communicable diseases such as diabetes and hypertension).

Adaptations to improve linkage to care among those diagnosed centered around telemedicine or community-based support to reach those not engaging in care. Recognizing and supporting healthcare provider and patient needs during the COVID-19 pandemic was important, and some adaptations focused on empowering patients and providing social (e.g., food parcels) or financial support. During the COVID-19 pandemic and the lockdown, private-public partnerships aided the innovation and advancement in technology. For example, to meet the demand for telecommunication services and wider connectivity both sectors partnered to improve fiber connectivity, so much so that the number of households and businesses in South Africa with fiber connections has risen by a staggering 4,200% in 2023 [26]. Digital tools and virtual technologies were rapidly adopted by medical officers in hospitals to monitor key indicators and provide technical support to teams virtually–minimizing exposure and protecting healthcare workers [27].

Adaptations to promote retention in care focused on different ways to deliver medication and also support patients on treatment. At the beginning of the pandemic treatment practices

were modified to reduce visits to health facilities and minimize the risks of COVID-19 exposure. Multi-month dispensing and decanting patients to external pick-up points (including private pharmacies) closer to home were some of the adaptations implemented to reduce frequent visits to the facility and free up resources. Healthcare utilization in South Africa also fell during COVID-19 and the lockdown as people were unable to access health services for a variety of reasons (e.g., fear, stigma, lack of transport, restrictions on movement, etc. (Loveday et al., 2020). Therefore, facilities implemented measures to deliver services to the household or in the community. Virtual models using telephonic (calls, texts, short message services) or video-directly observed treatment (DOT) were used to support adherence.

## Discussion

This study examined how TB and HIV services were adapted during the COVID-19 pandemic in South Africa. In South Africa, adaptations primarily focused on enhancing screening, testing, and diagnosis, improving medication collection and delivery, and adopting virtual models. The latter two themes were also among the most commonly implemented worldwide. As with other research, this study identified significant risks to the control of HIV and TB due to the pandemic [28]. Some of the main factors that contributed to service disruptions included: reductions in the number of health facilities offering services, lack of human and financial resources, redirection of HIV and TB services towards COVID-19, disruption in transportation of medication and laboratory supplies, and widespread limitations on freedom of movement and loss of wages for people to get to health facilities [29].

Our study concurs with other reports which have highlighted the benefits of healthcare service measures implemented during the pandemic. These measures include 1) innovative pharmaceutical dispensing methods; 2) innovative strategies to create awareness, spread health information, and promote engagement in health services; 3) telehealth to provide patient care; and 4) task shifting to redistribute healthcare workloads [30]. These measures have the potential to strengthen health systems and decongest facilities, reducing the frequency of contact between clients and healthcare providers, which may still be of value and have long-term benefits post-pandemic.

### Integrating HIV and TB services into COVID-19 protocols

Despite these significant barriers, adaptations were made in healthcare services, particularly with regard to the integration of HIV and TB services into COVID-19 testing protocols. As one study notes, in South Africa, there was a massive refocusing on the underlying social determinants that impact HIV and TB health outcomes during the pandemic (i.e., poverty, overcrowding, undernutrition, poor quality housing, etc.) [31]. In particular, these authors describe how efforts were made to enhance TB and HIV services in the context of COVID-19, such as by developing community networks for COVID-19 screening which incorporated TB and HIV screening, in addition to an expansion of TB and HIV self-testing. As mentioned above, the benefits of this include bringing services closer to patients, making accessing services more convenient, reducing the frequency of visits to the health facility, and reducing patient costs associated with health visits; thereby making services more people-centered. Another study corroborates these goals by noting how point-of-care diagnostic tests for COVID-19 antigen detection provided a unique opportunity to strengthen community testing and early detection of HIV and TB [29]. Additional research from Johannesburg, South Africa, highlighted how simple and feasible models can integrate TB testing into COVID-19 care pathways at primary care facilities [32]. The authors emphasized that combining operational workflows for TB and COVID-19 can lead to lasting integration of testing for multiple illnesses.

Of the interventions included in our analysis, over a third of the interventions targeted TB service delivery. This is hardly surprising as TB and COVID-19 affect the lungs and are transmitted mainly by aerosols or particles of saliva from infected persons so these similarities presented an opportunity for bi-directional screening and testing for TB and COVID-19, improving overall case detection for two diseases, and optimizing already constrained resources. Results from a WHO survey which spread across 32 countries (21 in the WHO AFRO region) reported that dual testing from TB and SARS-CoV-2 in patients who could have both or either disease was the most frequently implemented strategy. Molecular testing using the GeneXpert platform (Xpert MTB/RIF Ultra cartridge for TB and Xpert Xpress SARS-CoV-2 cartridge for SARS-CoV-2) was also reported as the primary method used for TB and SARS-CoV-2 testing [33]. Bi-directional screening and testing of COVID-19 and TB leverage the synergy of the two programmes and save resources as the same health worker and the same testing platform is used.

## Multi-month drug distribution

Next, many studies highlighted the importance of providing multi-month medication distributions to people living with illnesses such as HIV and TB in the context of the COVID-19 pandemic and attendant problems with routine healthcare provision. For example, nine sub-Saharan African countries including South Africa made all patients eligible for multi-month distribution from the initiation of ART during the pandemic [34]. As these authors describe, recent trial data from five sub-Saharan African countries, including South Africa, corroborate the superiority of multi-month medication distributions compared with shorter ART refills. Other studies that examined data from throughout sub-Saharan Africa confirm these findings, arguing that HIV programs that include the provision of several months of ART medications instead of one or two months showed improved health outcomes in the context of the pandemic [35, 36]. In addition to providing multi-month distribution of medications for HIV, research has confirmed the utility of similar approaches for TB, with programs allowing multi-month dispensing of drugs, allowing family members or friends to collect anti-TB drugs for patients, and reducing the frequency of outpatient visits for treatment monitoring [29].

## Medication delivery

Apart from enabling patients to acquire multiple months' worth of medication in one go, medication delivery emerged as another strategy for HIV and TB patients, significantly enhancing adherence and health outcomes both in sub-Saharan Africa and globally. One study that examined home delivery of ART in Indonesia, Laos, Nepal, and Nigeria found that in all four countries ART home delivery was a feasible and acceptable approach for ensuring HIV treatment adherence during COVID-19 lockdowns and travel restrictions [37]. The authors found that home delivery models in these countries were rapidly designed and successfully implemented to meet the emergency needs of patients and should be implemented at a greater scale in the future. In sub-Saharan Africa, eleven countries, including South Africa, were found to emphasize community-based models for ART delivery during COVID-19 [34]. This study argued that innovative strategies for different contexts allowed patients to easily access medications from private pharmacies, and identify external pick-up points in their communities, as well as lockers or "pele boxes" as they were referred to in South Africa. In South Africa, innovative treatment delivery strategies such as these allowed many patients on HIV and TB medication to continue to manage their conditions, and have since been expanded for other chronic conditions [29]. Another study found that delivery models–in addition to multi-month dispensing–were particularly beneficial for young people in South Africa [38].

## Telemedicine

Lastly, telemedicine was increasingly utilized as a strategy during the pandemic to ensure patient access to doctors and other healthcare professionals. In addition to allowing doctors to continue to provide advice and support for patients, these platforms permitted healthcare professionals to provide care to those living with HIV and TB and reduce exposure to COVID-19 [29]. While particularly successful in South Africa due to higher rates of access to technology and internet connectivity, telemedicine achieved some success elsewhere in sub-Saharan Africa as well [35]. However, those without internet or video capabilities were also able to have clinical or counseling consultations over the telephone–particularly those who are initiating treatment or re-engaging with services [31]. These authors describe how the decentralization of services through mobile technologies and telemedicine provides an opportunity for true service delivery transformation that can result in lasting benefits for the care of people living with HIV and TB. A recent systematic review, which included 24 studies on virtual healthcare services and digital health technologies adopted during COVID-19 in South Africa, revealed that these technologies had been used for screening of infectious and non-infectious diseases, disease surveillance and monitoring, medication and treatment compliance, and creating awareness and communication. However, the review highlighted that the use of smart technologies was not without limitations. These limitations included infrastructural and technological barriers, organizational and financial constraints, policy and regulatory hurdles, as well as cultural barriers [27]. As Lalla-Edwards and colleagues [20] highlight in their study, the government needs to formulate health intervention strategies that embrace health literacy, alternate methods of chronic medication dispensation, improved communication across healthcare platforms, and the use of telehealth, to avoid the threats of possible further infectious disease outbreaks.

## Maintenance of programs post-pandemic

As the COVID-19 deaths eased, restrictions were lifted and HIV and TB programs gradually improved, suggesting that disruptions to the treatment of individuals with HIV and TB were temporary and due to the pandemic [29]. However, many of these innovations including integration of HIV and TB to other services, multi-month dispensing, medication delivery, and telemedicine can serve as measures for expanding access to care not only when health services are disrupted but also under routine circumstances [37]. In South Africa in particular, these relatively new adaptations should be leveraged by existing systems as targeted interventions to improve health outcomes.

## Limitations

There were several limitations to the studies examined for this research. Firstly, we searched only one database (MEDLINE) and extracted data exclusively from English-language papers. However, since the rapid review was intended to inform the key intervention groups/themes, we deemed this approach sufficient, especially after reaching saturation, where no new themes emerged. Most interventions focused on screening, testing, diagnosis, or medication delivery and collection. Fewer focused on adapting clinic visits, awareness, patient education, monitoring and evaluation, as well as reporting. Additionally, few interventions had comparison groups, the majority were small in scale, single site, with limited follow-up or no outcomes reported. For virtual models such as video-directly observed therapy (DOT), it is difficult to evaluate implementation fidelity as medication adherence cannot be monitored remotely.

In the absence of data on feasibility, acceptability, and cost-effectiveness it is difficult to make any recommendations regarding which interventions should be integrated as part of the standard of care. Cost-effectiveness data is needed to decide which interventions are

sustainable and should be adopted as part of better service delivery. Lastly, few reported patient or provider experiences or perspectives, and, likely, some interventions were not documented. Also, some of the interventions may no longer be appropriate following the global decline of COVID-19 cases, relaxed government restrictions, and South Africa's vaccine strategy released in January 2021. We tried to repeat the e-survey 12 months after the first e-survey (mid-2023) to ascertain how many organizations were still using the interventions. Unfortunately, due to a poor response, this data could not be included.

## Conclusion

The COVID-19 pandemic demonstrated that services can be delivered in locations other than solely in health facilities and with fewer facility-based interactions. It is, therefore, key to strengthen community-based care and scale up self-care models, which can help overcome some of the barriers to health seeking for TB, HIV, and other chronic conditions that were observed during COVID-19. Additionally, integrated services are well received by patients and can free up additional providers and resources. Virtual models not only assist in patient management but also empower healthcare workers and facilitate the monitoring and evaluation of TB and HIV programs. The experiences and lessons from this study can help expand efforts to get TB and HIV care back on track following the COVID-19 pandemic and help build a more resilient healthcare system.

## Supporting information

**S1 Table. Summary of data fields extracted and grouped by main intervention theme (March 2020 and December 2021).**
(DOCX)

**S2 Table. E-survey.**
(DOCX)

**S1 Data. Full study data.**
(XLSX)

## Acknowledgments

We would like to thank Isla Kuhn from Cambridge University Medical Library, School of Clinical Medicine, University of Cambridge, Cambridge for her help the search strategy for the literature review.

## Author Contributions

**Conceptualization:** Denise Evans, Aneesa Moolla.

**Data curation:** Lezanie Coetzee, Vongani Maluleke, Patricia Leshabana.

**Writing – original draft:** Michael Galvin.

**Writing – review & editing:** Jacqui Miot.

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
