## [Decision Letter · Decision Letter 0]

22 Jul 2024

PGPH-D-24-00762

Adopting sustainable innovations for remote access to TB and HIV Care in South Africa

Dear Dr. Galvin,

Thank you for submitting your manuscript to PLOS Global Public Health. After careful consideration, we feel that it has merit but does not fully meet PLOS Global Public Health’s publication criteria as it currently stands. Therefore, we invite you to submit a revised version of the manuscript that addresses the points raised during the review process.

We look forward to receiving your revised manuscript.

Kind regards,

Lei Gao

Academic Editor

Journal Requirements:

1. We note that your Data Availability Statement is currently as follows: We have provided the raw data as a "Supplementary File". Data can also be available upon request from the authors or from the Health Economic & Epidemiology Research Office (HE2RO) at the University of the Witwatersrand which can be reached at information@heroza.org.

2. Please provide separate figure files in .tif or .eps format.

Additional Editor Comments (if provided):

Reviewers' comments:

Reviewer's Responses to Questions

**Comments to the Author**

1. Does this manuscript meet PLOS Global Public Health’s publication criteria? Is the manuscript technically sound, and do the data support the conclusions? The manuscript must describe methodologically and ethically rigorous research with conclusions that are appropriately drawn based on the data presented.

Reviewer #1: Yes

Reviewer #2: Yes

Reviewer #3: Yes

2. Has the statistical analysis been performed appropriately and rigorously?

Reviewer #1: Yes

Reviewer #2: No

Reviewer #3: Yes

3. Have the authors made all data underlying the findings in their manuscript fully available (please refer to the Data Availability Statement at the start of the manuscript PDF file)?

Reviewer #1: Yes

Reviewer #2: No

Reviewer #3: Yes

4. Is the manuscript presented in an intelligible fashion and written in standard English?

Reviewer #1: Yes

Reviewer #2: Yes

Reviewer #3: Yes

5. Review Comments to the Author

Reviewer #1: A very good paper and quite informative. It would be very helpful if the same study could be reproduced if other databases other than MEDLINE were to be used .Furthermore feasibility studies on the effectiveness, sustainability and cost effectiveness would help guide on how best the ICTS can be best integrated in existing programs particularly TB and HIV programs.

Reviewer #2: An interesting study but it must be -organized a bit further with more rigorous data documentation including actual data disaggregation with charts and tables of the literature review and interventions. Making the distinction between programmatic and differentiated service delivery interventions and providing greater insight on the human resource capacity challenges would be helpful.

Reviewer #3: Coming out of the COVID-19 pandemic, it is crucial to publish high-quality review articles to stimulate progress in advancing major initiatives on dual TB and HIV epidemics. The article's premise offers an opportunity.

This is a well-written manuscript that purports to focus on the importance of implementing several initiatives to adapt the service delivery model in HIV and TB in response to the COVID-19 pandemic. A detailed analysis of these initiatives globally focused on the S. African experience is presented, followed by an exciting discussion about practical implications for the country to maintain and extend best practices for integrating HIV and TB services to other services, medication delivery, and telemedicine after the COVID 19 pandemic, highlighting the importance of additional cost-effectiveness data for sustainability of selected interventions.

I have a few comments as follows:

1. Abstract; Rows 69 - 70: Suggest elaborating on the meaning of “services can be delivered outside of health facilities”. I also believe that the paragraph between rows 69 to 75 needs to be more aligned with the key points presented in the discussion section, such as the maintenance of programs post-pandemic.

2. Row 244: Suggest elaborating on the meaning of the "changes" that have been made in response to challenges.

3. Row 513: Linked with the comment on the abstract above, I suggest providing more details about the types of services that can be offered outside of health facilities.

6. PLOS authors have the option to publish the peer review history of their article (what does this mean?). If published, this will include your full peer review and any attached files.

**Do you want your identity to be public for this peer review?** For information about this choice, including consent withdrawal, please see our Privacy Policy.

Reviewer #1: No

Reviewer #2: No

Reviewer #3: No

---

## [Decision Letter · Decision Letter 1]

23 Aug 2024

PGPH-D-24-00762R1

Adopting sustainable innovations for remote access to TB and HIV Care in South Africa

Dear Dr. Galvin,

Thank you for submitting your manuscript to PLOS Global Public Health. After careful consideration, we feel that it has merit but does not fully meet PLOS Global Public Health’s publication criteria as it currently stands. Therefore, we invite you to submit a revised version of the manuscript that addresses the points raised during the review process.

We look forward to receiving your revised manuscript.

Kind regards,

Lei Gao

Academic Editor

Journal Requirements:

Additional Editor Comments (if provided):

Reviewers' comments:

Reviewer's Responses to Questions

**Comments to the Author**

1. If the authors have adequately addressed your comments raised in a previous round of review and you feel that this manuscript is now acceptable for publication, you may indicate that here to bypass the “Comments to the Author” section, enter your conflict of interest statement in the “Confidential to Editor” section, and submit your "Accept" recommendation.

Reviewer #2: All comments have been addressed

Reviewer #3: All comments have been addressed

2. Does this manuscript meet PLOS Global Public Health’s publication criteria? Is the manuscript technically sound, and do the data support the conclusions? The manuscript must describe methodologically and ethically rigorous research with conclusions that are appropriately drawn based on the data presented.

Reviewer #2: Yes

Reviewer #3: Yes

3. Has the statistical analysis been performed appropriately and rigorously?

Reviewer #2: Yes

Reviewer #3: Yes

4. Have the authors made all data underlying the findings in their manuscript fully available (please refer to the Data Availability Statement at the start of the manuscript PDF file)?

Reviewer #2: Yes

Reviewer #3: Yes

5. Is the manuscript presented in an intelligible fashion and written in standard English?

Reviewer #2: Yes

Reviewer #3: Yes

6. Review Comments to the Author

Reviewer #2: Abstract:

• Perhaps emphasize what can be learned from TB focused DSD services from this study.

• Could there have been a bias for English language studies that were only reviewed in the PRISMA checklists give the different non English countries on the WHO high TB Burden list. E.g Russia, Mozambique.

Results/South Africa

• In terms of implementation the total for district, provincial and facility level do not add up to 100% (5/57 Missing) could the rest be unknown

• What were the gaps between the gaps observed in TB/HIV integration in the local setting compared to the global setting in the literature review. This could have been summarized highlighted further in the discussion as seen in Figure 1

• Cost effectiveness of the innovative interventions would have been nice to compare the innovation versus existing routine costs for TB/HIV services and is noted in the limitations

• Outlining the various successful innovations e.g telemedicine, Medication delivery, multi-month dispensing was very effective.

Conclusion

• One could have stressed more the importance of virtual based platforms to improve the quality of TB/HIV service delivery as highlighted in the narrative discussion and results.

Reviewer #3: (No Response)

7. PLOS authors have the option to publish the peer review history of their article (what does this mean?). If published, this will include your full peer review and any attached files.

**Do you want your identity to be public for this peer review?** For information about this choice, including consent withdrawal, please see our Privacy Policy.

Reviewer #2: **Yes: **Dr. Cleophas d'Auvergne

Reviewer #3: No

---

## [Editor Report · Decision Letter 2]

12 Sep 2024

Adopting sustainable innovations for remote access to TB and HIV Care in South Africa

PGPH-D-24-00762R2

Dear Dr. Galvin,

We are pleased to inform you that your manuscript 'Adopting sustainable innovations for remote access to TB and HIV Care in South Africa' has been provisionally accepted for publication in PLOS Global Public Health.

Best regards,

Lei Gao

Academic Editor